# Inhibitory Action of Antidiabetic Drugs on the Free Radical Production by the Rod Outer Segment Ectopic Aerobic Metabolism

**DOI:** 10.3390/antiox9111133

**Published:** 2020-11-15

**Authors:** Silvia Ravera, Federico Caicci, Paolo Degan, Davide Maggi, Lucia Manni, Alessandra Puddu, Massimo Nicolò, Carlo E. Traverso, Isabella Panfoli

**Affiliations:** 1Dipartimento di Medicina Sperimentale, Università di Genoa, Via De Toni 14, 16132 Genova, Italy; silvia.ravera@unige.it; 2Dipartimento di Biologia, Università di Padova, via U. Bassi 58/B, 35121 Padova, Italy; federico.caicci@unipd.it (F.C.); lucia.manni@bio.unipd.it (L.M.); 3U.O. Mutagenesis and Preventive Oncology, IRCCS Ospedale Policlinico San Martino, L.go R. Benzi, 10, 16132 Genova, Italy; paolo.degan@hsanmartino.it; 4Department of Internal Medicine and Medical Specialties, University of Genova, 16132 Genova, Italy; davide.maggi@unige.it (D.M.); alep100@hotmail.com (A.P.); 5Clinica Oculistica (DINOGMI), Università di Genova, V.le Benedetto XV 6, 16132 Genova, Italy; massimo.nicolo@unige.it (M.N.); mc8620@mclink.it (C.E.T.); 6Fondazione per la Macula onlus, Università di Genova, V.le Benedetto XV 6, 16132 Genova, Italy; 7Dipartimento di Farmacia (DIFAR), Università di Genova, V.le Benedetto XV 3, 16132 Genova, Italy

**Keywords:** F_o_F_1_-ATP synthase, glybenclamide, diabetic retinopathy, light, metformin, oxidative stress, reactive oxygen species and intermediates, rod outer segment

## Abstract

Rod outer segments (OS) express the F_o_F_1_-ATP synthase and the respiratory chain, conducting an ectopic aerobic metabolism that produces free radicals in vitro. Diabetic retinopathy, a leading cause of vision loss, is associated with oxidative stress in the outer retina. Since metformin and glibenclamide, two anti-type 2 diabetes drugs, target the respiratory complexes, we studied the effect of these two drugs, individually or in association, on the free radical production in purified bovine rod OS. ATP synthesis, oxygen consumption, and oxidative stress production were assayed by luminometry, oximetry and flow cytometry, respectively. The expression of F_o_F_1_-ATP synthase was studied by immunogold electron microscopy. Metformin had a hormetic effect on the OS complex I and ATP synthetic activities, being stimulatory at concentrations below 1 mM, and inhibitory above. Glibenclamide inhibited complexes I and III, as well as ATP production in a concentration-dependent manner. Maximal concentrations of both drugs inhibited the ROI production by the light-exposed OS. Data, consistent with the delaying effect of these drugs on the onset of diabetic retinopathy, suggest that a combination of the two drugs at the beginning of the treatment might reduce the oxidative stress production helping the endogenous antioxidant defences in avoiding retinal damage.

## 1. Introduction

Genetic and environmental factors are involved in the pathogenesis of Type 2 diabetes mellitus (T2D), which is characterized by high blood glucose levels and insulin resistance in target organs [1]. In T2D, hyperglycemia and unbalanced oxidative stress determine both macro- and micro-vascular damage, bearing multifactorial pathogenesis. Insulin resistance increases mitochondrial superoxide production, promoting inflammation and triggering biochemical pathways associated with T2D complications, such as polyol metabolic pathway, advanced glycation end products formation, eNOS, and prostacyclin synthase inhibition) [2]. These, in turn, promote reactive oxygen intermediates (ROI) production. 

The most common ocular T2D complication is diabetic retinopathy (DR) [3,4], characterized by changes in the retinal microcirculation and neuronal damage. Establishing which of the two is the primum movens has proven difficult. The most prominent DR presentation is vascular. However, recently, it has been observed that retinal neurons are also damaged in the early stages of DR [5]. In fact, a significant percentage of newly diagnosed T2D patients display some degree of retinopathy [5]. Neurodegeneration is an early manifestation of DR associated with oxidative stress [6]. Recently, it was demonstrated that oxidative stress in early DR is restricted to photoreceptors [7]. Oxidative stress plays a central role also in other retinal degenerations, including age-related macular degeneration (AMD) [8], generally characterized by a photoreceptor and retinal pigmented epithelium (RPE) oxidative damage.

The electron transfer chain (ETC) is the principal source of ROI formation [9,10,11] as dysfunctional oxidative phosphorylation (OxPhos) increases the likelihood of superoxide generation. Research has mainly focused on anomalies in retinal mitochondrial function. However, experimental data have also shown that the outer segments (OS) disks and other unrelated membranes express F_o_F_1_-ATP synthase (ATP synthase) and ETC, conducting an ectopic aerobic metabolism (reviewed in [12,13]). The expression of the mitochondrial OxPhos machinery in the OS [14,15,16,17] would account for the high O_2_ consumption of the outer retina [3]. A correlation has been reported among oxidative stress correlated to the aerobic metabolism and the light absorption in isolated rod OS, showing for the first time that the phototransduction requires an increase in ATP production through O_2_ consumption in the OS, which are devoid of mitochondria [18]. This supports the existence of a potential source of oxidative stress in the rod OS disks that contain a high percentage of polyunsaturated fatty acids (PUFA) [10,19]. Notably, ROI production appeared higher in the OS than the inner segment (IS) after the irradiation with short-wavelength blue light (BL) [19].

T2D management encompasses lifestyle education programs and glycemic regulation by the use of oral antidiabetic agents. These increase insulin secretion from the pancreatic β cells, reduce insulin resistance as well as the postprandial glycemia [20]. Metformin (MTF) is a biguanide, recommended as first-line therapy of T2D in non-pregnant adults by the American Diabetes Association (ADA) and the European Association for the Study of Diabetes (EASD) [21,22]. MTF allows good glycemic control with minimum hypoglycemic risk. In particular, MTF stimulates the AMP kinase (AMPK) pathway, improving insulin sensitivity in muscle and adipose tissue, and inhibits the glucose-6-phosphate phosphatase expression, suppressing the hepatic gluconeogenesis [23]. MTF has also anti-inflammatory, and caloric restriction-related antiaging activity [24] consistent with the effect that we have previously described in the rod OS [18]. Moreover, MTF displays an inhibitory effect on the complex I of the ETC [25]. It was shown that MTF plays a protective role for both photoreceptors and RPE [20,26]. MTF can also be utilized in combination with sulfonylureas such as glibenclamide (GLB), also known as glyburide in the US. Interestingly, GLB also targets the mitochondrial respiratory complexes I, II, and III, but not IV, determining a reduction of oxygen consumption and ATP synthesis in H9c2 cardiomyoblast cells [27,28]. GLB also caused a dose-dependent increment of the AMP/ATP ratio, due to a deregulated energy balance similar to that obtained with MTF.

The purpose of this work was to study how MTF, GLB or a combination of the two drugs affects the production of ROI and the aerobic metabolism of the OS purified from bovine retinas in vitro. 

## 2. Materials and Methods 

### 2.1. Materials

All chemical compounds were of the highest chemical grade. Metformin (MTF) and glibenclamide (GLB) were purchased from Sigma-Aldrich (S. Louis, MO, USA).

### 2.2. Purified Bovine Rod OS Preparations

All operations were carried out under dim red light at 4 °C. Eye semi-cups were obtained from 24 freshly enucleated bovine eyes obtained from a local certified slaughterhouse in Ceva (Cuneo, Italy). After the anterior chamber and vitreous removal, retinas were collected after inducing their free-floating by filling the eye semi-cups with Mammalian Ringer (MR; composed of 0.157 M NaCl, 5 mM KCl, 0.5 mM MgCl_2_, 8 mM NaH_2_PO_4_, 7 mM Na_2_HPO_4_ and 2 mM CaCl_2_ pH 6.9) plus protease inhibitor cocktail (Sigma-Aldrich, S. Louis, MO, USA) and 50 µg/mL ampicillin and incubating for 10 min [29]. Then rod outer segments (OS) were purified following the method of Schnetkamp and Daemen [30]. The lower band in the continuous Ficoll/sucrose gradient, corresponding to the sealed isolated OS was collected, diluted with two volumes of 600 mM sucrose, 200 mM Tris-HCl pH 7.4, centrifuged at 5000× *g* for 20 min at 4 °C and stored at −80 °C [31]. Before use, OS aliquots were thawed and homogenized by diluting the suspensions 1:4 (*v*/*v*) in ultrapure water (Milli-Q^®,^ Millipore, Billerica, MA, USA), and subjecting them to at least 10 passages through a needle (25 gauge) on ice [31]. The purity of the OS preparations was assessed characterizing them for the integrity of the plasma membrane as reported [30,32] by assaying the protein content of the supernatant, after centrifugation of samples at 20,000× *g* for 2 min. Rhodopsin (Rh) concentration, as a specific OS marker was determined spectrophotometrically by measuring the difference in absorption spectra at 500 nm recorded before and after exhaustive bleaching (light, for 5 min) of samples (0.06 mg/mL, in ultrapure water, with a dual-beam spectrometer (UNICAM UV2; Analytical S.n.c., Parma, Italy), using a molar extinction coefficient of 41,000 cm^−1^ M^−1.^ Rh concentration in the original OS preparation was around 0.7 mM.

### 2.3. Transmission Electron Microscopy 

In order to localize the ATP synthase β-subunit, immunogold transmission electron microscopy (TEM) was performed as described [31,33]. Bovine eye semi-cups were dissected and fixed with 4% paraformaldehyde and 0.1% glutaraldehyde in PBS buffer. After fixation (ON at 4 °C.), pieces of fixed cups were dehydrated, embedded in LR White Resin, and polymerized at 58 °C. Specimens were then cut, using an ultramicrotome; sections about 90 nm thick were placed on nickel grids and used for post-embedding immunogold experiments. Sections on grids were treated with blocking solution (1% BSA, 0.1% Tween 20, PBS), then with mouse monoclonal anti-rhodopsin (1:100) (Sigma Aldrich, St. Louis, MO, USA) and rabbit polyclonal anti-ATP synthase β-subunit (diluted 1:50) (Sigma–Aldrich) overnight at 4 °C. For antibody binding detection, secondary goat anti-mouse IgG (Sigma Aldrich) (diluted 1:100) coupled to gold particles (40 nm), and goat anti-rabbit IgG (Sigma Aldrich) (diluted 1:100) coupled to gold particles (10 nm) were used. Controls were performed by omitting primary Ab, which resulted in absence of cross-reactivity (data not shown). Grids were analyzed at a FEI Tecnai G^2^ transmission electron microscope operating at 100 KV. In negative controls, the pre-immune serum was applied to the sections instead of the specific primary Ab. Images were acquired with OSIS Veleta cameras, collected and typeset in Corel Draw X8.

### 2.4. Antidiabetic Drugs Treatment on Rod OS

MTF and GLB were resuspended in ultrapure water or DMSO, respectively. Rod OS were incubated with GLB (25, 50, 100, 200 µM) or with MTF (15 µM, 150 µM, 1.5 mM, 2.5 mM, 5 mM) for 1 h in the dark at 25 °C. For the combination experiments, the samples were incubated with 5 mM MTF + 100 or 200 µM GLB, or with 150 µM MTF + 25 or 50 µM GLB. Afterward, aliquots of suspension were employed for the biochemical analyses. 

### 2.5. Respiratory Complex I Activity Assay

Complex I (NADH-ubiquinone oxidoreductase) activity was measured spectrophotometrically at 420 nm, following the reduction of ferricyanide (ε_mM_ for FeCN^−^ =1.0 M^−1^ cm^−1^). For each assay, 0.035 mg/mL of total protein were used. The assay solution contained: 0.6 mM NADH, 0.8 mM ferricyanide, 50 mM Tris-HCl pH 7.4 (TRIS pH 7.4), 50 mM KCl, 5 mM MgCl_2_, 1 mM EGTA, and 50 µM Antimycin A [14,31].

### 2.6. Respiratory Complex III Activity Assay

Complex III (cytochrome *c* reductase) activity was evaluated spectrophotometrically at 550 nm, following the reduction of oxidized cytochrome c (Cyt c, ε_mM_ for Cyt *c* = 20 M^−1^⋅cm^−1^), employing 0.035 mg/mL of total protein. The assay medium contained: 0.03% oxidized Cyt *c*, 0.6 mM NADH, 20 mM succinate, 50 mM TRIS pH 7.4, 5 mM KCl, 2 mM MgCl_2_, and 0.5 M NaCN [14,31].

### 2.7. ATP Synthesis Assay 

An amount of 20 µg of total OS homogenate protein was incubated for 5 min at 37 °C in an assay solution composed by: 50 mM Tris-HCl pH 7.4, 50 mM KCl, 1 mM EGTA, 2 mM MgCl_2_, 0.6 mM ouabain, 0.25 mM di(adenosine)-5-penta-phosphate (Ap5A, adenylate kinase inhibitor), and 25µg/mL ampicillin (0.1 mL final volume). As respiratory substrates, 5 mM pyruvate plus 2.5 mM malate were added to the incubation medium. ATP synthesis was induced by the addition of 5 mM KH_2_PO_4_ and 0.2 mM ADP, at the same pH of the mixture. ATP formation was followed for 2 min in a luminometer (Lumi-Scint, Bioscan, Washington, D.C. USA) by the luciferin/luciferase chemiluminescent method (Roche Diagnostics Corp., Indianapolis, IN, USA). The calibration curve was obtained with ATP standard solutions (Roche Diagnostics Corp., Indianapolis, IN, USA) from 10^−9^ and 10^−7^ M in the same solution of the experiments [14,31].

### 2.8. Cytofluorimetric Assays

Reactive oxygen intermediates (ROI) production by the purified OS homogenates (1.8 mg/mL) (40 µg of total protein/0.6 mL) was analyzed by flow cytometry, carried out using an ADA CyAn cytometer (Beckman Coulter, Brea, CA, USA) equipped with three laser lamps, using the fluorescent probe dihydrorodamine 123 (DHR) (Molecular Probes, Life Technologies, Carlsbad, CA, USA) as described [18,31]. Aliquots of the OS homogenates were first incubated for one hour at 25 °C with GLB (25, 50, 100, 200 μM) or with MTF (15 μM, 150 μM, 1.5 mM, 2.5 mM, 5 mM) dissolved in DMSO or ultrapure water, respectively. In other experiments, the combination of the two drugs has been employed. Incubation with only DMSO was made as control of GLB treatment experiments. Treated samples were resuspended in 500 μL of the following solution: 10 mM HEPES, 135 mM NaCl, 5 mM CaCl_2_. Then, DHR (2.5 mg/mL) and respiring substrates (0.2 mM NADH and 10 mM succinate) and 0.1 mM ADP were added to the suspension. Samples were kept in ambient light to elicit the light-dependent development of ROI.

### 2.9. Statistical Analysis

Statistical significance was tested by the analysis of variance (ANOVA) for multiple comparisons, using the GraphPad Prism version 5.00 statistical software (GraphPad Software Inc., La Jolla, CA, USA). Values of *p* < 0.05 were considered significant [31].

## 3. Results

### 3.1. F_1_F_o_-ATP Syhthase is Expressed in Bovine Retinal Mitochondria as well as Rod Outer Segments

The Transmission Electron Microscopy (TEM) immunogold analysis confirms the localization of ATP synthase in the OS of the bovine retinal rod [15,31] (Figure 1). In detail, Panel B shows that the signal of rhodopsin (Rh; 40 nm diameter gold particles) colocalizes with the signal of β-subunit of ATP synthase (10 nm diameter gold particles) in rod OS disks. Conversely, rod IS mitochondria display only the ATP synthase signal (Panel C).

### 3.2. Metformin Displays a Hormetic Effect on Aerobic Metabolism and Oxidative Stress Production, While Glibenclamide Inhibits the Oxidative Metabolism

To evaluate the effect of metformin (MTF) and glibenclamide (GLB) on the rod OS extramitochondrial aerobic metabolism, the activity of the respiratory complexes I and III, and of ATP synthase were evaluated in the presence of the antidiabetic drugs, compared to controls in which only the vehicles (ultrapure water or DMSO) were present. As reported in Figure 1, pre-treatment of the OS with MTF displays a hormetic effect on complex I (Figure 2A): it induces an enhancement of activity at low concentrations (15 and 150 μM), while determining a drastic reduction at higher concentrations (1.5, 2, or 5 mM). A similar trend was observed on the ATP synthesis (Figure 2C), confirming that complex I activity in rod OS is linked to energy production [14]. By contrast, MTF does not affect the activity of complex III either at low or at high concentrations (Figure 2B).

Since oxidative metabolism is always associated with oxidative stress [34], we have also assayed the ROI production in rod OS suspended in a glucose medium and exposed to light. In this way, the ETC is strongly activated, and the production of ROI sensibly enhanced [18]. It was observed that the therapeutic concentrations of MTF (15 and 150 μM) determine an increment, while the higher MTF concentrations induce a reduction of ROI production (Figure 2D). Conversely, pre-treatment of the OS with all the employed GLB concentrations (25, 50, 100, and 200 μM) displays a negative effect on both the aerobic metabolism and the ROI production. In particular, both complex I and complex III appear affected by GLB, determining a significant reduction of both the ATP synthesis and the associated oxidative stress production (Figure 2).

### 3.3. The Combination of High Concentrations of Metformin and Glibenclamide Determines a Drastic Reduction of Aerobic Metabolism and Relative Oxidative Stress Production

The literature reports in vitro studies using high MTF concentrations to inhibit the OxPhos activity in several cell types, including tumors [25,35]. Since GLB shows a similar behavior on complex I activity but also affects the complex III activity [27], we evaluated whether a combination of these anti-diabetic drugs can exert additive effects on the aerobic metabolism of rod OS. Data show that the combination of MTF and GLB determines a significant additive inhibitory effect on complex I activity, in comparison to the single treatment with GLB. However, the inhibitory effect of GLB is weaker in comparison to that induced by MTF (Figure 3A). The double treatment exhibited stronger inhibitory activity also on the ATP synthase activity compared with the decrement obtained with single treatments (Figure 3B). This is probably due to the combined additive effect on complex I plus the inhibition of complex III activity.

This inhibitory effect is also exerted on the production of free radicals, which appear to be further reduced by the combination compared to the single treatments (Figure 3C). 

### 3.4. The Metformin-Dependent Increment of Energy Metabolism and Oxidative Stress is Reduced by the Addition of Low Concentrations of Glibenclamide

As shown in Figure 2, and as reported in the literature [36], the therapeutic doses of MTF induce an increment of the oxidative metabolism and the related oxidative stress production, which can negatively affect the OS and the retina. Therefore, we asked whether the combination of the therapeutic doses of MTF with low GLB concentrations, which showed a mild inhibitory effect on energy metabolism, could reverse such increase.

Data show that the activity of complex I in the combination condition is significantly lower than in samples untreated or treated with low MTF doses (Figure 4A). By contrast, complex I activity was higher compared to single treatment with GLB. The same trend is observed for the ATP synthesis and ROI production (Figure 4B,C), suggesting that GLB can mitigate the enhancement of aerobic metabolism and oxidative stress induced by the low MTF doses.

## 4. Discussion

Diabetic retinopathy (DR) is one of the complications of T2D and represents the principal cause of blindness in adults [37]. Many metabolic abnormalities have been associated with the development of DR, including an unbalanced local oxidative stress in the outer retina [7,38,39,40]. In particular, rod OS appear vulnerable to oxidative stress [8,17,41,42], which drives the retinal degeneration [17,18]. On the other hand, antioxidants were shown to mitigate signs of DR in animal diabetic models [43] and in cultured photoreceptor cells (661W) [44]. 

Oxidative stress is mainly due to the mitochondrial aerobic metabolism [34], and, in particular, is correlated to complex I activity [45]. However, the oxidative stress in the outer retina may not solely depend on the mitochondrial OxPhos, but also on an extramitochondrial aerobic metabolism [14] functionally expressed in the OS [15,18], as confirmed by the TEM immunogold analysis reported in Figure 1, showing the ectopic expression of ATP synthase in the OS. Such aerobic energy production in the rod OS would sustain the ATP/GTP supply for the phototransduction, together with the IS mitochondria [16,46]. However, the OxPhos is a major producer of oxygen radicals, with the ETC being the principal ROI producer in the cell [47]. Indeed, samples come from healthy bovine retinas, but the OS is a suitable model to evaluate the effect of modulators of the ETC, such as MTF and GLB, on the oxidative stress produced by the aerobic metabolism. In fact, our positive controls were samples exposed to light, in order to hyper activate the OS ectopic ETC, functionally linked to phototransduction [18], and enhance ROI production [31]. This phenomenon likely occurs, in vitro as the OS have lost the dioptric media of the eye and preparation has diluted the physiological concentrations of antioxidants. Supposing that such free radical production can occur also in vivo in pathological conditions, the extramitochondrial rod OS metabolism could be the main contributor to the oxidative stress production in the OS, which is the recognized basis of the DR pathogenesis.

Both MTF and GLB, largely used in type 2 diabetes therapy, target the respiratory complexes. This work aims to evaluate the effect of these two drugs, in a single treatment or in combination, on the activity of respiratory complexes I and III, ATP synthase, and on the ROI production in the rod OS. The OS appears to be a suitable experimental model, as the OS disks display a similar sidedness of the inverted mitochondrial vesicles. Moreover, OS homogenates in vitro produce ROI upon exposure to light, acting as a “molecular switch” [18], as the aerobic OS metabolism is linked to phototransduction. In addition, the oxidative damage of OS has been related to sunlight and blue light exposure [19,48] and to the elevated PUFA content of the OS disks [49]. Notably, 4-hydroxynonenal (4-HNE), a lipid peroxidation product, is found primarily in the OS [19,50]. 

MTF is used as the first choice for T2D patients since it reduces gluconeogenesis and improves glucose uptake and utilization [24]. Recently, relatively high MTF concentrations have been found to also display an effect on aging and cancer [24,51], as well as in oxidative stress response [52], since doses greater than 1 mM result in inhibition of the respiratory complex I [25]. However, the in vivo effects of MTF remain not fully understood [53]. Nonetheless, GLB exerts an action on aerobic metabolism, by inhibiting the respiratory complexes I, II, and III [27].

Our data show that MTF displays a hormetic behavior: doses similar to the plasma drug concentration in diabetic patients (15–150 μM) [25,35,54] increase OxPhos activity and an increment of ROI production. By contrast, high concentrations (1.5, 2.5, 5 mM) inhibit the activity of complex I and the associated aerobic metabolism, as already previously reported [18]. Such a hormetic effect of MTF in vitro is confirmed by previous results on H9c2 cells [27]. Since the modulator effects of MTF on OS aerobic metabolism were observed after a very short incubation time, it is possible to speculate that these effects are due at least partly to a direct action on the molecular machinery of the OxPhos. On the other hand, in hepatocyte cultures, the positive effect of low MTF concentration on the aerobic metabolism seems correlated to AMPK activation, which would promote mitochondrial fission in order to improve mitochondrial respiration [55]. Since the positive effect on MTF in hepatocytes is observed after a few hours of treatment, it may be speculated that the same mechanism occurred in our experimental conditions. Accordingly, AMPK exerts a neuroprotective effect in case of rod and cone inflammation [56].

The aerobic OS metabolism is also negatively affected by GLB, which acts on both complex I and complex III activities [27]. Although the inhibitory effect mediated by GLB on complex I activity appears lower in comparison to that of MTF, it is important to note that GLB also inhibits complex III, inducing a complete blockade of the OxPhos pathway. This effect was reflected in the ATP synthetic activity, similar to what observed with MTF.

When the high MTF dose (5 mM) was combined with the high GLB doses (100 or 200 μM), we observed a strong decrement of OxPhos activity and ROI production, suggesting an additive effect between the two drugs. In particular, the double inhibitory effect on complex I and the negative regulation of complex III by their combination drastically reduced the ETC activity, reflected on both the ATP synthetic ability and the ROI production. 

The most interesting data are the inhibitory effect of low GLB doses on the increase in OxPhos activity observed with plasma doses of MTF (150 μM). In fact, while the single treatment with 150 μM MTF induces an increment of OxPhos metabolism and related ROI production, the combination with low doses of GLB (25 or 50 μM) determines a significant reduction of respiratory complexes activity, the ATP synthesis, and oxidative stress production in comparison to the single MTF low-dose treatment. This suggests that the combination of the two antidiabetic drugs could avoid the increment of oxidative damage triggered by MTF, while maintaining the aerobic metabolism at a good level, thus ensuring enough energy supply to support phototransduction. 

However, our data do not wish to suggest that MTF therapeutic dosages would induce retinal damage due to increased local oxidative metabolism. By contrast, the literature reports that MTF plays a protective role on the photoreceptors and the RPE [26]. For example, according to a retrospective study there is a correlation between the long-term MTF treatment and the reduced severity of DR in patients with T2D, regardless of their HbA1c level [57]. Long-term use of metformin was independently associated with a significantly lower rate of severe non-proliferative or proliferative DR [57] and significantly lower risk of development of AMD [58]. This apparent discrepancy could depend on two factors. Firstly, it can be supposed that the natural antioxidant systems would be sufficient to overcome this transient damage due to the increased OxPhos activity associated with the MTF treatment, even though a relative deficiency in reduced glutathione was observed in the OS, explaining its vulnerability to oxidation [59]. On the other hand, the oxidative damage does not depend only on an increase in the production of oxidative stress, but also on the unbalance between ROI production and antioxidant response [39]. Secondly, the beneficial effect of MTF therapy on glycemic control would decrease inflammation levels and the relative oxidative stress production, independently of glucose catabolism.

Therefore, from a clinical point of view, our data allow us to speculate that in diabetic patients without a severe DR, the administration of MTF could be sufficient to avoid retinal damage. Conversely, the combination of GLB and MTF could be useful for those patients bearing advanced DR, to improve the metabolic state of the OS while at the same time avoiding the increment in local oxidative stress production. Further studies are needed to confirm these hypotheses.

We have previously shown that modulation of ATP synthase by polyphenols and terpenes sensibly lowers ROI production by light-exposed OS [18,31]. Such an action is beneficial until the oxidative damage uncouples the respiring disks: then, inhibition of the ectopic ATP synthase would not stop ROI production anymore. By contrast, modulation of the ETC complexes with substances able to inhibit some of the complexes may protect from oxidative damage interrupting the vicious cycle causing metabolic dysfunction and photoreceptor apoptosis. This scenario may not be remote, since, in the central nervous system, photoreceptors consume more oxygen (about 4 times) than the other tissues do, on a weight basis [60]. Measurements of the oxygen tension in human subjects with DR showed high oxygen tensions at the posterior pole, absent in controls [49]. The present data also suggest that the RPE would be an innocent bystander damaged by trafficking the oxidized cargo represented by the oxidatively damaged OS disks. It was proposed that changes in the subretinal space hydration, as measured by optical coherence tomography, represent its oxidation state [43], which is impaired by oxidative stress in DR patients.

## 5. Conclusions

The results reported here add new data on the action of GLB and MTF as inhibitors of the ETC, in a model of OxPhos. Data confirm that the rod OS ectopic ETC is a source of cytosolic oxidative stress in saturating light in vitro, which is prone to generating oxidative damage. Free radical production linked to phototransduction may occur also in vivo, under dysmetabolic and membrane damage conditions such as those occurring in the pathogenesis of DR.

## Figures and Tables

**Figure 1 antioxidants-09-01133-f001:**
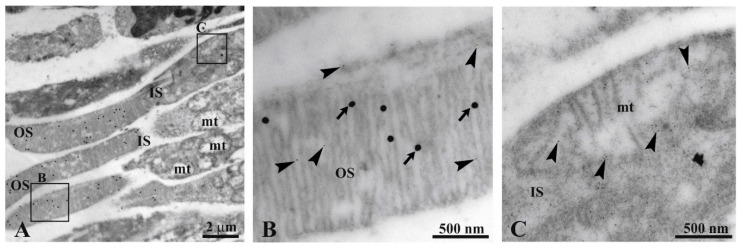
Immunogold experiment with transmission electron microscopy (TEM) on bovine retina. (**A**) Retina section showing inner (IS) and outer (OS) photoreceptor segments. (**B**,**C**) Enlargements corresponding to the squared areas B, C in Panel A, to show a detail of an OS and mitochondrion. Largest gold particles (40 nm width, arrows) reveal Ab against anti-rhodopsin in OS. The smallest gold particle (10 nm width, arrowhead) reveals Ab against anti-ATP synthase β-subunit in both OS and a mitochondrion.

**Figure 2 antioxidants-09-01133-f002:**
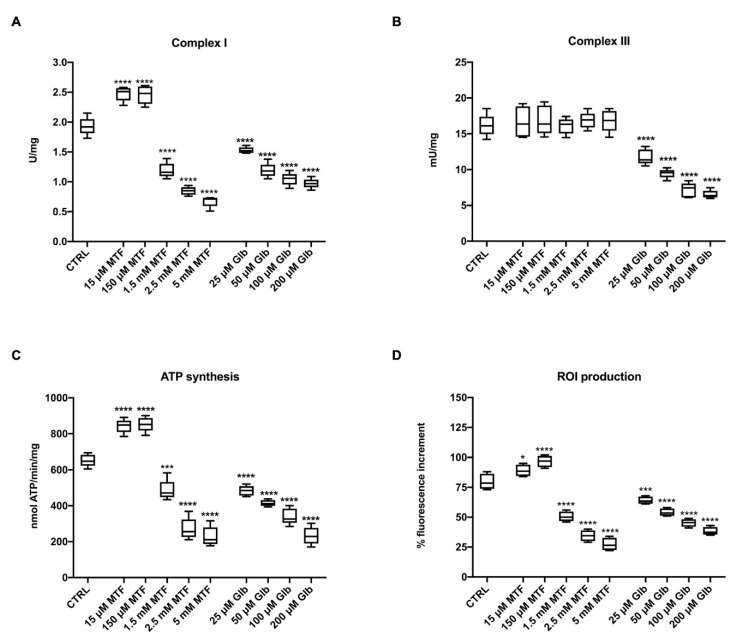
Effect of metformin (MTF) and glibenclamide (GLB) on the aerobic metabolism and reactive oxygen intermediates (ROI) production in rod OS; the figure reports the effect of 15 μM, 150 μM, 1.5 mM, 2.5 mM, 5 mM MTF and 25, 50, 100, 200 μM GLB on the aerobic metabolism in rod OS. (**A**) Complex I activity, expressed as U/mg of total protein. (**B**) Complex III activity, expressed as mU/mg of total protein. (**C**) ATP synthesis trough F_o_-F_1_ ATP synthase expressed as nmol of ATP produced/min/mg of total protein. (**D**) ROI production in rod OS, exposed to light. Data are from *n* = 5 independent experiments; *, ***, or **** indicate a significant difference for *p* < 0.05, 0.001, or 0.0001, respectively, between treated and untreated samples (CTRL).

**Figure 3 antioxidants-09-01133-f003:**
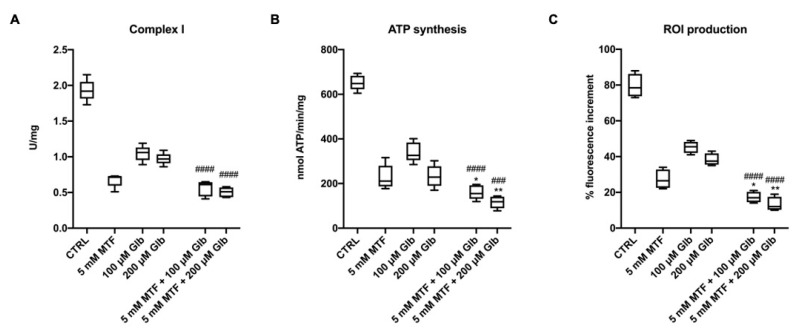
Effect of combination treatment with high doses of MTF and GLB on the aerobic metabolism and ROI production in rod OS. Figure reports the effect of the combination treatment compared with the single treatment. The used treatments are: 5 mM MTF + 100 μM GLB and 5 mM MTF + 200 μM GLB. (**A**) Complex I activity, expressed as uU/mg of total protein. (**B**) ATP synthesis trough F_o_–F_1_ ATP synthase expressed as nmol of ATP produced/min/mg of total protein. (**C**) ROI production in rod OS exposed to the light. Data are from *n* = 5 independent experiments. * or ** indicate a significant difference for *p* < 0.05 or 0.01, respectively, between the combination treatment and 5 mM MTF treatment. ### or #### indicate a significant difference for *p* < 0.001 or 0.0001, respectively, between the combination treatment and the respective exclusive GLB treatment.

**Figure 4 antioxidants-09-01133-f004:**
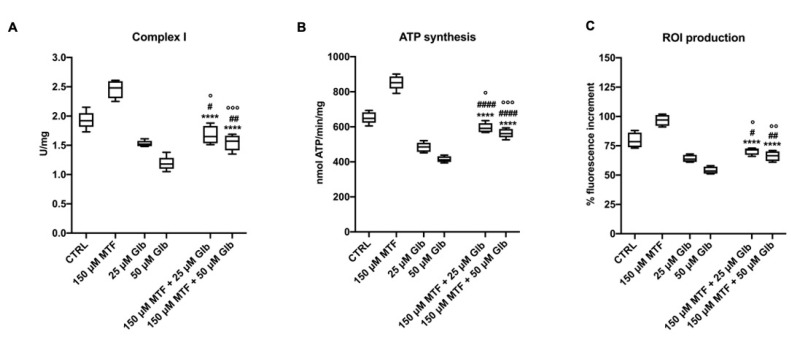
Effect of combination treatment with low doses of MTF and GLB on the aerobic metabolism and ROI production in rod OS. Figure reports the effect of the combination treatment compared with the single treatment. Treatments were: 150 μM MTF + 25 μM GLB and 150 μM MTF + 50 μM GLB. (**A**) Complex I activity, expressed as uU/mg of total protein. (**B**) ATP synthesis trough F_o_–F_1_ ATP synthase expressed as nmol of ATP produced/min/mg of total protein. (**C**) ROI production in rod OS exposed to the light. Data are from *n* = 5 independent experiments. **** indicates a significant difference for *p* < 0.0001 between the combination treatments and 150 μM MTF treatment. #, ## or #### indicate a significant difference for *p* < 0.05, 0.01 or 0.0001, respectively, between the combination treatment and the respective only GLB treatment. °, °° or °°° indicate a significant difference for *p* < 0.05, 0.01 or 0.001, respectively, between the combination treatment and the untreated samples.

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
