# Peer review of "Inhibitory Action of Antidiabetic Drugs on the Free Radical Production by the Rod Outer Segment Ectopic Aerobic Metabolism"

_antioxidants, 2020, doi:10.3390/antiox9111133_

Round 1
Reviewer 1 Report
The manuscript explores mechanisms involved in DR pathogenesis. While I can follow the authors explanations I feel there are a few points which could benefit the reader to appreciate their results:
- the experiments assess bovine samples which are free from DR...so a bit more discussion as to how this translates into DR pathogenesis beyond the processes that are involved is necessary
- the discussion should offer more detail on the clinical impact of the results presented.
Lastly, the manuscript would benefit from English editing:
- the vast amount of sentences are very long.....and in most cases could be split into 2 or more.
- wording: line 257: "a massive"..this is jargon and should be removed
- line 259: "degenerations" should be replaced by "degeneration"
- Line 262: "only" should be replaced by "solely"
- line 271: "sideness"....I'm not sure what is referred to here?
- These are just a few examples and careful reading through the manuscript should help identify these spelling/ grammar issues
Author Response
Reviewer 1
The manuscript explores mechanisms involved in DR pathogenesis. While I can follow the authors explanations, I feel there are a few points which could benefit the reader to appreciate their results:
We thank the Reviewer for his/her appreciation.
- The experiments assess bovine samples, which are free from DR...so a bit more discussion as to how this translates into DR pathogenesis beyond the processes that are involved is necessary
We thank the Reviewer for this suggestion.
Indeed, the samples used come from healthy bovine retinas, not affected by DR. However, we have been utilizing the purified rod OS as a suitable model to study both the oxidative phosphorylation and the photoreceptor oxidative damage: (i) the OS disk membranes display an inside out sidedness, like mitochondrial inverted vesicles and (ii) the OS metabolism can be activated by light up to the point to produce ROI, in vitro: when the purified OS, in the absence of the dioptric media of the eye and after dilution of the physiological concentration of antioxidants are exposed to light, they produce a considerable amount of ROI, likely due to the activity of the ETC, a major free radical producer [1]. Therefore, aerobic ectopic metabolism in the rod OS is inextricably linked to phototransduction whose over-functioning increases the demand of GTP (ATP), with an over-functioning of the ETC. We have supposed that such free radical production occurs also in vivo under pathological conditions, as briefly already reported in the first part of the discussion: the extra-mitochondrial rod OS metabolism could be a major contributor to the oxidative stress production in the rod OS, which is the recognized basis of the DR pathogenesis. In this respect, the RPE would be an innocent bystander, in that it would suffer from phagocytizing oxidized disk membranes [2].
Therefore, evaluating the effect of various concentrations of MTF and GLB on the ectopic rod OS OxPhos, we may underline the contribution of the unsuspected aerobic metabolism of the rod OS to the pathogenesis of DR. This part was added to the discussion in the revised version of the manuscript.
- the discussion should offer more detail on the clinical impact of the results presented.
We thank the Reviewer for the suggestion. In the revised version, we have discussed more deeply the clinical implications of our results, although further studies are needed to confirm our assumptions.
Lastly, the manuscript would benefit from English editing:
- the vast amount of sentences are very long.....and in most cases could be split into 2 or more.
We apologize for the poor English language. In the revised version, we have checked and corrected the length of sentences.
- wording: line 257: "a massive”. This is jargon and should be removed
We apologize for this mistake. The word has been removed in the revised version.
- line 259: "degenerations" should be replaced by "degeneration"
We apologize for this mistake. The word has been corrected.
- Line 262: "only" should be replaced by "solely"
Thank for the suggestion. The word has been replaced.
- line 271: "sidedness"....I'm not sure what is referred to here?
We apologize for the mistake. The sentence has been rephrased.
- These are just a few examples and careful reading through the manuscript should help identify these spelling/ grammar issues.
We thank the Reviewer for these suggestions. In the revised version, the English language has been checked and corrected.
Reviewer 2 Report
This manuscript was focused on two DM drugs or their combination on mitochondrial functions of rod OS. The methods were OS isolation, EM, Complex I/II assay, ATP synthesis and ROS production. The whole story was straightforward. However, some issues need to be clarified.
1.The "dose" should be changed as "concentration" in Abstract (line 31, p1).
2. Please describe where the slaughterhouse located (line 96, p3).
3. How to evaluate the purity of rod OS preparation?
4. What was the vehicle of these two drugs used?
5. Is there any reagent for the positive control to "activate" respiratory Complex I ?
6. Since sclareol used as a f1f0-ATP synthase inhibitor (2020), did it used as a positive inhibitor (line 154, p11) in this experiment? These findings of scareol should be described in the results.
7. What is the inhibitory selectivity of sclareol between rod OS- and other tissue-mitochondria?
8. Please describe "combo" as "combination" in the legend of Fig. 3.
9. The description of line186 (p5) should be moved and discussed in the Discussion section.
10. The subtitle of "3.4. The addition ....of metformin" need to be revised (line 231, p6).
11. Please discuss the process of "light" induce mitochondria-related ROI in OS. Is there another mitochondria-independent process to produce ROI in photorecepters?
12. The paragraph of lines 278-283 (p11) or lines 297-301 (p11) need to be revised or merged as possible.
13. "The oxygen consumption" of low concentration of Met should be clarified for its Complex I activation and ROI overproduction in OS.
14. English grammar need to be revised.
Round 2
Reviewer 2 Report
The revised manuscript was well done. The descriptions of rod OS purity were mentioned in detail. The issues of sclareol were also carefully explained in response. Especially, the discussion of mitochondria-independent ROS was well included. The final issue is to revise English grammar of the Abstract.